# The Luminos Project: Co-Designing a Short-Stay Suicide Support Model for Young People

**DOI:** 10.3390/ijerph22091449

**Published:** 2025-09-18

**Authors:** Aims Hansen, Samantha Speirs, Kirsten Panton, Jacinta Freeman, Zrinka Highfield, Kieren Marshall, Eleanor Tighe, Laura Hemming, Bep Uink, Francis Mitrou, Vu Vuong, Ashleigh Lin

**Affiliations:** 1School of Population and Global Health, The University of Western Australia, 35 Stirling Hwy, Nedlands, WA 6009, Australia; aims.hansen@uwa.edu.au (A.H.); samantha.speirs@uwa.edu.au (S.S.); 2School of Psychological Science, The University of Western Australia, 35 Stirling Hwy, Crawley, WA 6009, Australia; kirsten@thesamaritans.org.au; 3The Samaritans, 60 Bagot Road, Subiaco, WA 6008, Australia; zrinka@thesamaritans.org.au; 4The Kids Research Institute Australia, 5 Hospital Avenue, Nedlands, WA 6009, Australia; jacinta.freeman@thekids.org.au; 5Ruah Community Services, L3-4/29 Shenton Street, Northbridge, WA 6003, Australia; kieren.marshall@ruah.org.au (K.M.); ellie.tighe@ruah.org.au (E.T.); francis.mitrou@thekids.org.au (F.M.); vu.vuong@thekids.org.au (V.V.); 6Violet Vines Marshman Centre for Rural Health Research, La Trobe Rural Health School, La Trobe University, Bendigo, VIC 3550, Australia; l.hemming@latrobe.edu.au; 7Health InfoNet, Edith Cowan University, 2 Bradford St, Mount Lawley, WA 6050, Australia; b.uink@ecu.edu.au; 8Centre for Child Health Research, The University of Western Australia, 35 Stirling Hwy, Crawley, WA 6009, Australia

**Keywords:** suicide prevention, youth, relational models

## Abstract

*Background:* Suicide was the leading cause of death among young Australians aged 15–24 years old in 2023, with 392 lives lost. The continued high numbers of youth suicide demand urgent exploration of alternative approaches to suicide intervention in this population. The United Kingdom-based suicide service Maytree offers an innovative short-term stay for people experiencing suicidal thoughts. Grounded by the Maytree model-of-care, the aim of the current study was to co-design a short-stay service responsive to the specific needs of suicidal young people. *Methods:* Semi-structured and focus group interviews with young people (*n* = 38), caregivers (*n* = 11) and key local stakeholders (*n* = 26) in Perth, Western Australia. *Results:* Deductive and inductive thematic analysis identified 8 core themes: benefits, service design, staffing, operations, referrals, challenges and safety, measures of success, and language. Endorsement of the Luminos model as beneficial to young people experiencing thoughts of suicide was nearly unanimous. *Conclusions:* These findings provide actionable insights for the development of alternative, youth-informed suicide support services.

## 1. Introduction

In Australia, suicide is the leading cause of death for people aged 15–24 years, with 298 lives lost in 2023 [1].

Over the past decade, there has been a significant and concerning increase in youth suicide rates, reaching 31.8% among 15–17-year-olds and 33.1% among 18–24-year-olds in 2023. These figures were notably lower in 2021—at 16.5% and 23.9%, respectively—underscoring the urgent need for innovative and effective approaches to suicide prevention and support [1]. For those who are experiencing suicidal ideation, hospital Emergency Departments (ED) are often the first services to be engaged. However, many young people find that EDs fall short in addressing their mental health crises. They report long wait times, insufficient privacy and a lack of support after discharge [2], with many reporting it is unhelpful and often not culturally safe. They also report negative interactions with staff, including experiences of stigma, feeling dismissed, and being treated as an inconvenience [2].

In recent years there has been a shift away from medical and risk-averse models of care in favor of a more client-oriented approach [3,4,5,6]. Consultations with individuals who have lived or living experience with suicide consistently highlight the importance of a client-led approach, which involves a trusted therapeutic relationship tailored to individual needs and preferences [6,7]. However, there remains a gap in knowledge on the effectiveness of alternative forms of suicide intervention [6].

There is growing recognition that conventional, medicalized responses to suicide often fail to provide the kind of support that people in distress find meaningful. These models tend to focus on individual risk, diagnosis, and crisis containment, often at the expense of connection, dignity, and long-term recovery [8]. In contrast, relational and non-clinical approaches are gaining traction for their emphasis on presence, trust, and human connection. Rather than viewing suicidality solely through a medical or risk management lens, emerging approaches such as critical suicidology reframe it as a relational, social, and context-dependent experience. This field contributes to the broader shift by challenging dominant biomedical frameworks, centering lived experience, and questioning the narrow focus on risk and pathology. It advocates for care models that promote agency, meaning, and mutual support—aligning more closely with how many individuals experiencing suicidality understand and seek help [8].

Aligned with these emerging frameworks, residential peer-support services offer an important alternative to conventional clinical care. These services provide short-term, home-like environments where individuals can access relational, peer-led emotional support outside of clinical settings, often without needing to meet the criteria for acute hospital admission.

An example of this is the Maytree Model, a UK-based service providing non-clinical, short-term residential support for people experiencing suicidal distress [9]. Designed for adults, Maytree offers a four-night, five-day stay in a home-like environment that prioritizes emotional support over risk management. Grounded in a relational model of care based on openness and befriending, Maytree aims to reduce stigma and affirm suicidality as part of the broader human experience. An evaluation of the service found significant improvements in mental health by the end of the stay, with these improvements continuing in the months that followed [9]. Using the Clinical Outcomes in Routine Evaluation (CORE) measure, the study showed that average scores across emotional well-being, psychological symptoms, daily functioning, and risk-to-self dropped sharply from arrival to discharge and continued to improve at 3–6-month follow-up period. By this point, most guests’ scores had fallen below the clinical threshold for concern, suggesting that most guests no longer experienced distress levels requiring clinical intervention [9].

The relational, peer-led principles exemplified by the Maytree model have been reflected in comparable initiatives, such as Australia’s Suicide Prevention and Recovery Centre (SPARC) [10]. Like Maytree, SPARC provided a non-clinical, short-term residential service offering emotional respite and support to individuals experiencing suicidal distress outside of acute crisis. Developed through co-design with people with lived experience, clinicians, and advocacy groups, SPARC integrated peer and mental health support workers to foster relational care, agency, and empowerment. Evaluation results from SPARC mirrored those reported for Maytree, showing clinically significant reductions in psychological distress, improved functioning, and decreased perceived risk. Participants valued the informal, non-treatment-based relationships with peer workers, which contributed to enhanced social connectedness and increased confidence to engage with further supports [10].

Building on the relational, non-clinical principles exemplified by Maytree and reflected in SPARC, this study focuses on developing a youth-specific suicide respite model. Young people face distinct developmental, cultural, and systemic barriers to accessing effective suicide support, and their help-seeking preferences often differ from adults [2]. Consequently, there is a clear need for services tailored to their unique needs. This study aims to adapt and incorporate foundational principles—such as peer support, emotional connection, and trauma-informed care—into a model designed specifically for young people in Western Australia. Through co-design consultations with young people, caregivers, and stakeholders in Perth, it identifies core elements that define an effective youth-centered, trauma-informed suicide respite service. The findings provide an evidence-based framework to guide the creation of a feasible, responsive alternative to conventional clinical interventions, supporting more meaningful and accessible suicide support for young people.

## 2. Methods

### 2.1. Participants

Seventy-five participants were recruited for the co-design phase of *The Luminos Project*. Young people (aged 16–24 years, *n* = 38) were eligible to participate if they had lived or living experience of suicidal thoughts and/or behaviors. Parents and carers (*n* = 11) were eligible if they were supporting, or had supported, a young person with lived experience of suicide. Stakeholders (*n* = 26) were eligible to participate if they worked with young people with suicidal thoughts and/or behaviors. Demographic information of the young people can be seen in Table 1.

Participants were recruited through social media advertisements, word of mouth and through organizational contacts. Participants were excluded from participating if they were actively suicidal with intent or a plan (as assessed by the clinical judgement of the research team), appeared to be under the influence of alcohol or other drugs at the time of the focus group or interview, or were unable to provide informed consent. All participants provided written informed consent, either electronically via Qualtrics or in person. Young people aged 16–17 years were consented as mature minors. Participants under the age of 16 required written consent from a parent or guardian. Young people and carers were offered AUD $50 reimbursement for their participation. Data collection occurred from September to November 2022.

### 2.2. Procedure

Participants completed either an interview or focus group, according to their preference. Overall, twelve young people, five parents/carers and five stakeholders completed interviews. Interviews ranged between 32 and 90 min. Twenty-six young people participated across five focus groups, six parents took part in one focus group, and twenty-one stakeholders participated across five focus groups. Focus groups were between 40 and 90 min. Interviews were completed either in person or online, and focus groups were completed in person. Interviews and focus groups were audio recorded and transcribed verbatim. Recordings were transcribed by Transcription Australia (https://www.transcription.net.au/ 26th of September to 7th of December 2022), whose platform is hosted on a 128-bit SSL secure connection to ensure confidentiality and data security. Ethics approval was obtained through the Western Australian Aboriginal Health Ethics Committee.

Separate interview guides were developed for each participant group, tailored to their specific experiences and insights. Group discussions were guided by a series of exploratory questions designed to inform the co-design of the youth suicide service model. These questions explored participants’ perspectives on service design, key features of care (including trauma-informed principles), roles and responsibilities of staff and volunteers, training and workforce requirements, referral and access pathways, length of stay and transitions, involvement of families and carers, and anticipated implementation challenges.

Cultural safety principles were embedded throughout the research design and engagement processes. Local Aboriginal community-controlled services were engaged as stakeholders, and a tailored interview guide was developed for Aboriginal participants to ensure that questions were contextually appropriate and respectful. All focus groups and interviews were conducted by experienced facilitators with research or clinical expertise in youth mental health, who had completed ASIST suicide intervention training and Aboriginal Cultural Awareness training. These measures were implemented to foster safe and inclusive spaces, supporting culturally responsive engagement in the co-design of the youth suicide service model.

### 2.3. Analysis

Thematic analysis was conducted following Braun and Clarke’s [11] six-phase framework: (1) data familiarization; (2) generation of initial codes; (3) identification of potential themes; (4) theme review; (5) defining and naming themes; (6) report production. Author LH began by engaging thoroughly with the data through repeated transcript readings to ensure comprehensive familiarity. Coding was undertaken in NVivo using a hybrid deductive–inductive approach. Deductive codes were derived from the pre-established categories embedded within the interview and focus group guides, providing an initial organizational framework. Concurrently, inductive codes capturing salient, emergent concepts within the data were developed. Initial coding and theme development were conducted by LH and subsequently reviewed and refined by SS, who had conducted majority of the interviews and focus groups and reviewed all transcripts for accuracy. Inductive codes were grouped into sub-themes, which were iteratively reviewed and integrated with the deductive framework to form a cohesive thematic structure that balanced participant-driven insights with the study’s predetermined focus areas.

Interviews were conducted with participants from each participant group until thematic saturation was reached. While the concept of saturation is debated in qualitative research [12,13], it is generally understood as the point at which additional data no longer yield novel insights or themes. In this study, saturation was reached when no further themes arose within each participant group. Ongoing, iterative analysis conducted alongside data collection allowed researchers to monitor the recurrence of ideas in line with the study’s aim.

## 3. Results

The results were formed from the topic guide used to prompt discussion in interviews/focus groups. These are presented in Figure 1.

### 3.1. Benefits

Participants almost unanimously agreed that The Luminos Project would be a beneficial service. Participants felt Luminos would be beneficial because it was community-based, engaging, age-appropriate, free and suicide-specific for young people. Participants liked that the service taught reflection and growth and provided more privacy for young people.

“*I think it’s really important to facilitate support services and accessible resources that are created as a preventive measure instead of a crisis management measure so that we instill a lot of skills and things like that for people to not have to get to the point of crisis*.”(Aboriginal young person 26)

Stakeholders and young people perceived Luminos to be a safer alternative than an ED or clinical services for suicidal young people.

“*I personally have been into a psych ward through the emergency department. It’s not a great experience so I understand why young people especially would want a space to actually go and deal with their mental health crisis*.”(Young person 16)

“*Once you’re going through mental health units, that’s often more traumatizing than the feelings of suicide and emptiness*…”(Stakeholder 4)

Most carers expressed that Luminos would offer respite for families. Young people spoke about respite for themselves from chaotic family households. For example:

“*[My ideal outcome would be] respite for the family knowing that their beloved is in an environment that’s safe for them... and… the individual needs of the child have been met*.”(Parent 1)

### 3.2. Design

All participants agreed Luminos should incorporate choice and flexibility into its design to empower young people and enhance comfort levels during their stay.

#### 3.2.1. Information Prior to Staying at Luminos

All participant groups highlighted the need for accessible, up-to-date information about Luminos to help young people determine if the service was right place for them. This included clear details on the scope of the service, policies on cultural safety (i.e., LGBTQA+ inclusion), confidentiality, technology use, substance use, as well as visual content of the residence (e.g., photos, videos, virtual walkthroughs) and testimonials to convey what staying at Luminos would be like. Preferences for communication formats varied; young people preferred social media, whilst stakeholders favored pre-stay visits and information sessions.

Participant groups wanted transparency around the application process, eligibility and waitlist times, as well as operational details such as rules, expectations, leave procedures, staff roles, activities, and approaches to managing risk and distress. Young people and carers frequently raised questions about food, including options, mealtimes, communal cooking and dining, and catering to dietary needs. Carers sought assurance that young people would be safe and should not be exposed to potential traumatic experiences. Stakeholders emphasized the need to explain the rationale behind the model of care, and ensure young people understood data collection and complaint processes.

#### 3.2.2. Physical Aspects

All participants consistently agreed that Luminos should be non-clinical and evoke a ‘warm’ and ‘homely’ atmosphere.

Layout: Preferences was largely aligned across groups, with participants supporting a mixture of shared and private areas. Commonly suggested shared areas included a kitchen, lounge, dining space, garden, crafts room, games room and a music room. Private spaces encompassed bedrooms, office spaces, bathrooms and quiet zones that could be used for privacy or reflection.

Bedrooms: A strong theme emerged across all participant groups that bedrooms should be private, not shared, to support young people’s comfort and autonomy. Young people placed particular emphasis on the ability to personalize their space, suggesting options such as different colors for each room, choice of décor (i.e., sheets, cushions, and pillows) and bringing comforting personal possessions and artwork. They also expressed a preference for gender-neutral, non-clinical design (e.g., no white sheets), adequate storage and space, and hidden mirrors to minimize triggers. Carers and stakeholders supported private rooms but were less focused on customization, instead emphasizing safety and comfort. The idea of lockable bedrooms divided opinion both within and across all groups, seen by some as essential for protecting belongings, while others viewed them as too reminiscent of clinical environments.

Bathrooms: Young people expressed a strong preference for private, lockable bathrooms, describing shared facilities as a potential barrier to engagement. Both young people and stakeholders raised concerns about the safety and trauma triggers associated with shared bathrooms, particularly the risk of an opposite gendered person entering whilst someone was changing. These concerns were especially pronounced for LGBTQA+ young people, with participants across groups emphasizing the need for gender-inclusive and non-triggering facilities. Carers were less explicit about bathroom preferences, though broadly supported the need for safety, privacy and inclusivity.

Shared areas: There was broad agreement across groups on the importance of shared spaces feeling comfortable and engaging. All participant groups suggested a variety of comfortable seating options (e.g., sofas, beanbags) and entertainment (e.g., TV, gaming console). Young people contributed more suggestions such as a pool table, arts and craft area, Bluetooth speakers, weighted blankets, stress balls and access to laundry facilities.

Furnishings: Participants across groups preferred furniture that felt “normal” and homelike rather than institutional. Young people and carers specifically emphasized the importance of high-quality, well-maintained furnishings to convey to young people they are valued and respected. Soft furnishings, such as cushions, blankets and carpeting, were commonly requested to enhance comfort and warmth.

Colors, décor and lighting: Consensus was not reached on a color scheme. All groups agreed that relaxing colors were necessary, however interpretation of relaxing colors varied. Some participants found bright colors were relaxing, while others found them overstimulating. There was consensus that colors should avoid resembling hospitals, with pastel or warm natural tones seen as most appropriate. Across groups, there was strong support for culturally inclusive décor, including Aboriginal and LGBTQA+ flags. Young people and stakeholders also valued plants, wall art and creative display areas such as whiteboards or chalkboards for personal artwork. All groups preferred warm and natural lighting, though young people uniquely proposed a low stimulation room (i.e., dim lighting, black out curtains) and dimmable lights to suit sensory needs.

Entrance: Participants agreed that the entrance to Luminos should feel calm and welcoming. Young people preferred a semi-private entry point to avoid walking directly into shared areas and expressed a desire for time alone in their room before joining others. Carers echoed the need for a non-intimidating entrance, raising concerns that overt security features could feel intimidating.

Outdoor space: Outdoor areas were valued across all groups, with common preferences for shaded areas, plants, peaceful aesthetics and cozy seating areas (e.g., hammock, swinging chair). Sporting equipment was recommended by all groups to promote movement and physical activity. Young people provided more detailed suggestions. Participants agreed on the need for a designated smoking area that was separate from the main outdoor space. Young people provided more detailed suggestions, such as a barbeque, a table, fairy lights, a pond, and a grass area, reflecting a desire for both relaxation and social interaction.

#### 3.2.3. Triggering Aspects of Physical Design

Participants across all groups agreed that specific objects, particularly those that could be used for self-harm, were more likely to be triggering than the overall design of the Luminos environment. Both stakeholders and young people emphasized the importance restricting access to items such as knives for safety. However, there were shared concerns that features associated with clinical settings, such as anti-ligature furniture and posted rules, could feel institutional and therefore triggering. Participants across all groups recommended that triggers be assessed on a case-by-case basis, with flexibility to make environmental adjustments to accommodate personal needs.

#### 3.2.4. Other Design Aspects

Young people and stakeholders agreed that Luminos should be centrally located and accessible by public transport. However, a minority of young people expressed a preference for a more rural location to avoid encountering peers. Both groups advocated for Luminos to have wheelchair accessibility and be child friendly to support young parents. Participants felt that temperature should be customizable.

#### 3.2.5. Rules

All participant groups agreed that rules were necessary to ensure safety but cautioned against an excessive or rigid approach to avoid replicating the feel of clinical settings. There was shared support for a collaborative process with young people staying in the development of rules with visible displays in the house. Young people emphasized the importance of understanding the rationale behind rules, as this would increase their willingness to follow them.

Young people and carers both advocated for clear, staged communication of rules: prior to arrival, at entry and throughout their stay. Carers proposed a welcome booklet, while young people favored a signed agreement, noting it would offer reassurance that expectations applied equally to all guests. Participants also agreed that staff should adopt a supportive rather than punitive approach to rule enforcement, focusing on the underlying causes of behavior. Young people further expressed the need for consistency in enforcement and assurance that serious breaches, such as discrimination against LGBTQA+ individuals, would be appropriately addressed.

#### 3.2.6. Alcohol, Illegal Drug Use and Smoking/Vaping

There was broad agreement across participant groups that Luminos should be free from alcohol and illicit drugs. Carers and stakeholders highlighted the challenge of enforcement, while young people emphasized that substance use should not be considered a barrier for entry into Luminos if the young person can abstain from using substances during their stay. All groups generally supported permitting smoking and vaping in a designated area, provided it did not affect others or interfere with participation. Young people stressed that banning smoking or vaping could remove a coping strategy.

#### 3.2.7. Mobile Phones and Technology

The use of mobile phones was a divisive issue within and across participant groups. Most young people supported unrestricted mobile phone use, viewing phones as a source of distraction and connection. However, a minority of young people, along with some carers and stakeholders, believed restrictions could be beneficial, particularly around sleep hygiene and reducing overstimulation. All groups agreed phones should not be used during scheduled activities and mealtimes. Opinions were split across and within groups regarding access to mobile phones at night. While some cited concerns about disrupted sleep and reduced engagement the next day, others believed phones could aid sleep or self-regulation. There was shared support for encouraging positive phone-use behaviors, such as contacting loved ones or listening to music/podcasts. Social media use was viewed more negatively by all groups, with young people specifically requesting boundaries around image sharing to protect privacy. Gaming was suggested by young people as a helpful unstructured and social activity. Carers and stakeholders did not raise strong objections but agreed with young people that limits should be in place to prevent interference with scheduled activities.

“*I think with phones, it’s not saying yes or no to a phone. I think it’s saying, ‘how are we using this? And how can we utilise this tool in a way that is helpful and not disruptive?’*”(Young person 3)

#### 3.2.8. Contact with Family and Friends

Concerns about contact with family and friends during a stay centered on whether relationships were genuinely safe and supportive. Carers and stakeholders highlighted how damaging or unstable relationships could interfere with a young person’s ability to reflect and heal. Young people acknowledged these risks but placed more weight on autonomy and the ability to choose who they stay connected with. Despite these differences, there was agreement that decisions about contact should be made on an individual basis.

#### 3.2.9. Unsupervised Leave

Support for unsupervised walks came primarily from young people and carers who saw them as beneficial for wellbeing. Stakeholders, by contrast, were more divided. Some backed limited unsupervised leave while others advocated for stricter oversight, including staff-accompanied walks, group outings or structured check-ins and curfews.

#### 3.2.10. Interactions and Engagement

Expectations around respectful and safe peer interactions were widely supported. Across groups, participants expressed the importance of boundaries, confidentiality and inclusive behavior. What separated young people’s perspectives was their strong preference that any emotionally heavy or potentially triggering conversations be kept between guests and staff, not shared among peers.

### 3.3. Staffing

#### 3.3.1. Staff Roles

All groups supported the proposed staffing structure, which included peer support workers, volunteers and a clinical lead. However, perspectives diverged on certain elements. Concerns about volunteer roles were raised across groups, with some participants questioning whether unpaid positions might result in lesser commitment or reliability. While young people and carers focused on the importance of staff being approachable and their roles clearly communicated, stakeholders placed stronger emphasis on integrated teamwork, warning against staff working in siloes. Stakeholders also proposed expanding the staffing model to include additional roles, such as skills coaches, Aboriginal Elders, social workers, occupational therapists, mental health nurses, chaplains and youth workers, suggesting a more multidisciplinary approach.

#### 3.3.2. Staff to Patient Ratio and Shift Pattern

Suggested staffing numbers ranged from two to five staff during the day, based on five guests staying; two to three was seen as the minimum and five as ideal. Opinions varied regarding decreased staffing at night. Some participants supported reducing staff numbers overnight, while others, particularly young people and carers, argued that nighttime is often a challenging and vulnerable time and adequate staffing was necessary. All groups requested overnight access to clinical support, whether in person or on-call. Young people and stakeholders emphasized the value of consistency in staffing during a stay, though young people noted that being paired with one worker for the entire stay could be challenging if the relationship did not work. Young people noted that shift changeovers should be timed carefully to avoid disrupting scheduled activities, an issue not raised by other groups.

#### 3.3.3. Staff Qualities

Figure 2 displays the qualities identified as important for staff to possess. The size of a word reflects how often a quality was mentioned. Key qualities include flexibility, adaptability, confidence, empathy, friendliness, kindness, open-mindedness, passion and relatability. Staff should not be condescending or authoritarian and should be able to listen well, be present and person-centered.

Workforce diversity was seen as essential by all participant groups, calling for representation from LGBTQA+ individuals, Aboriginal and Torres Strait Islander people, culturally and linguistically diverse communities, and alternative cultures (e.g., tattoos, piercings). Across groups, participants preferred staff to be of a similar or slightly older age than the guests but emphasized that the ability to understand and relate to young people outweighed age itself. An Aboriginal stakeholder added that the presence of Elders in the house could affect a young Aboriginal person’s ability to communicate with peers due to cultural protocols around respecting and engaging with Elders.

When describing the role of staff, young people focused on receiving emotional support, instilling hope, building self-confidence, and being given space to process their experiences while developing positive coping mechanisms. Carers and stakeholders shared many of these priorities but placed stronger emphasis on staff understanding young people’s language, accurately reading situations, and using their lived experiences in a positive and safe manner.

#### 3.3.4. Lived Experience in the Workforce

Although participants did not reach a consensus on the definition of “lived experience,” there was strong shared support for its inclusion in peer worker roles. Young people strongly valued staff members with lived experience related to suicide, seeing it as essential for fostering understanding and relatability. However, they also emphasized that personal qualities, such as empathy, were important and that not all staff needed to have lived experience themselves.

“*Not everyone has to have had mental health experience, or they might have a family member or someone else going through that…I just feel like if you can relate to them on some level, you can be buddies, I guess, and you feel like more comfortable with that*.”(Young person 27)

Self-disclosures made by peer workers about their personal experiences was viewed across groups as potentially beneficial if handled with care. Young people, carers and stakeholders all highlighted the importance of timing, relevance and avoidance of disclosing any potentially distressing details, particularly method-specific content. Young people tended to frame these disclosures as a way of fostering hope, however carers and stakeholders held concern about the potential for emotional harm.

“*I think I would just say around safely sharing… not talking potentially triggering things like methods or things in details but making sure they’re fostering hope*.”(Young person 31)

“*It might work, but then it might all end up being about them if they keep talking about it and it could be reliving the trauma*.”(Carer 11)

“*Certain ones might react really well to it, and others it might bring up emotions, it might enhance their feelings by listening to someone else. So, it’s getting a feel for who’s there first before you make that decision about talking about it*.”(Stakeholder 10)

#### 3.3.5. Recruitment, Training and Appearance

All participant groups endorsed the inclusion of young people on recruitment panels, with stakeholders additionally recommending the presence of an Aboriginal representative and a wellbeing assessment for potential staff. There was shared agreement on the importance of robust training, particularly in mental health first aid, crisis management, cultural awareness (Aboriginal and Torres Strait Islander, LGBTQA+, neurodiversity) and the appropriate use of self-disclosures. However, priorities varied across groups. Young people stressed the importance of training in areas directly affecting their experiences, such as service navigation, Auslan (Australian sign language), disability support and understanding the social determinants of health. Stakeholders focused more on risk assessment, coaching skills and recovery-orientated practice, reflecting a system-level perspective. Carers emphasized the need for training around medication delivery and safe administration.

Distinct group concerns also emerged. Stakeholders raised the need for ongoing emotional support for staff to prevent burnout and turnover, an issue not raised by other groups. Young people expressed preferences around staff presentation, including casual clothing and name badges displaying pronouns, reflecting their value on accessibility and inclusivity.

### 3.4. Operations

#### 3.4.1. Structure and Activities

Although participants agreed Luminos should have a clear structure, they emphasized the need for flexibility in scheduling to support young people’s autonomy. Downtime was viewed as essential to recovery, with suggestions for offering multiple structured activities on a non-compulsory basis. Activities were organized into four categories: structured or organized activities, unstructured activities, therapeutic activities and training activities, with several activities spanning more than one category (see Table 2).

#### 3.4.2. Meal Planning and Mealtimes

Across participant groups, there was agreement that meal planning, preparation and mealtimes should be collaborative. Stakeholders saw these activities as opportunities to teach nutrition and cooking skills. However, young people placed greater emphasis on flexibility to accommodate potential triggers (e.g., disordered eating, neurodiversity) and individual circumstances (e.g., dietary preferences, religious requirements). All groups, especially young people, agreed that nutritious food should always be available, not just during meals, to support autonomy and control over food intake. Young people additionally wanted a large kitchen for main meal preparation (overseen by a volunteer), a separate kitchenette for self-prepared snacks, and the option to bring their own food.

#### 3.4.3. Pets

Pets were seen as an important source of support for young people across all groups, with suggestions including organized sessions with trained emotional therapy animals or the possibility of Luminos having a house pet. Across groups it was noted that restrictions on pets in other residential services can act as a barrier to entry for young people unwilling to leave their pet behind. Some participants from each participant group suggested that young people could be able to bring their own pet, while others felt this could be triggering to other guests and should be assessed on a case-by-case basis.

#### 3.4.4. Rejection from Service

Participants across all groups raised concern that being declined from Luminos due to not meeting eligibility could be experienced as a personal rejection or a failure. Participants agreed that communication in these situations should be compassionate, sensitive and use clear language. Young people and stakeholders emphasized the importance of explaining the reasons for ineligibility and providing guidance on how eligibility could be met in the future. Participants across groups suggested offering referrals to other suitable services and, with consent, maintaining follow-up contact.

#### 3.4.5. Onboarding

Before a stay at Luminos, participants agreed key preparatory steps should include a risk assessment, development of a distress management plan, and collection of essential guest information. Within this, participant groups highlighted specific priorities. Young people emphasized the importance of understanding their individual needs, such as eating sensitivities, physical disabilities, or neurodivergent requirements. Carers prioritized a readiness assessment to confirm willingness and preparedness to engage, while stakeholders stressed the importance of having information about a young person’s triggers, previous trauma, and risky phone use. Across groups, there was consensus on supporting guests to maintain existing medical or counselling appointments during their stay, with these arrangements addressed in advance.

When discussing the welcome process, young people and carers stressed the value of being greeted by a friendly worker and avoiding an overload of information, paperwork, or activities upon arrival. Young people wanted an opportunity to share their preferences around routines and interactions (e.g., being woken up by staff), while stakeholders saw this as an opportune time to discuss goals for their stay. Young people noted feeling intimidated when introducing themselves to other guests and preferred introductions to be facilitated by staff. Carers suggested providing a welcome pack upon arrival containing key documents (e.g., weekly timetable, rules and regulations) and self-care items (e.g., soap) to help young people feel ‘seen and accepted’.

#### 3.4.6. Distress Management

Participants across all groups agreed a distress management plan should be created before entry, with young people consulted about their preferred regulation strategies when distressed. Learning to tolerate and manage distress was viewed as a valuable skill, and carers and stakeholders preferred that young people remain at Luminos during times of distress to practice in a safe and supportive environment. However, if distress levels became excessively high, participants felt it would be appropriate for the young person to leave Luminos and be supported to find a suitable safe place. Young people consistently emphasized the importance of staff explaining why emergency services were called, particularly given the cost of ambulances.

Young people suggested a range of supports to help manage distress, including de-escalation techniques (e.g., breathing exercises), and distraction methods to break unhelpful thinking patterns (e.g., going for a walk, watching TV, using ice cubes). They also suggested that staff should offer space, empathy, and validation, rather than resorting to immediate intervention.

#### 3.4.7. Staff Contact with Family and Others

All groups agreed that young people should control who their personal information is shared with and what details are disclosed. Young people wanted specific details of their stay to remain confidential but were open to inviting carers to actively participate in creating and implementing a safety plan for their transition out. Stakeholders emphasized the importance of providing updates to services to ensure a collaborative approach to care.

#### 3.4.8. Length and Timing of Stay

Participants, particularly stakeholders, felt the proposed length of stay (four nights, five days) was too short, especially for Aboriginal young people. Opinions differed on whether start dates should be staggered or simultaneous. Staggered starting would allow beds to be filled if someone left early but young people might feel anxious if they were the only newcomer among established guests. Simultaneous starts could support consistent relationship building but could potentially lead to personality clashes that would persist for the full stay. One suggestion was to consider personality types when scheduling intake. Aboriginal stakeholders additionally requested at least two Aboriginal young people staying at once to reduce feelings of isolation.

#### 3.4.9. Gendered Intake

Across groups, participants preferred a mix of genders but identified concerns such as the potential for past traumas to be triggered and the risk of sexual activity occurring. In response to these concerns, some stakeholders proposed gender segregated stays on alternating weeks, while one young person suggested LGBTQA+ specific intakes.

#### 3.4.10. Other Activity Related Concepts

Views varied across groups regarding activities and operational considerations. Carers and stakeholders felt young people should not engage in schoolwork or other tasks during their stay to maintain focus on recovery. Due to safety concerns about medications (e.g., overdose, theft by residents), participants suggested that only trained medical staff should handle medication dispensing. Young people and carers emphasized the need for private spaces where young people can participate in religious activities, such as prayer. Carers also expressed a need for direct support from Luminos for both them and the young person, including psychoeducation and therapeutic interventions.

### 3.5. Accessing and Exiting Luminos

#### 3.5.1. Referrals into Luminos

Across all groups, participants strongly agreed that the referral process should be straight forward and offer flexible pathways, including self-referrals and referrals by family. Young people wanted services to be able to make referrals on their behalf, while stakeholders suggested referrals should be completed collaboratively with young people. Stakeholders also requested a consistent point of contact for referral information. Young people wanted flexibility in how Luminos staff engage with them during the referral process, suggesting options such as phone calls, video conferencing, text messages or in-person meetings.

Suggestions for promoting Luminos varied across groups. Carers suggested holding open days where service providers could attend onsite information sessions. Stakeholders suggested sending regular emails to potential referrers to keep them engaged and noted that word-of-mouth from young people would be an important form of advertisement.

#### 3.5.2. Exiting Luminos

There was agreement across groups that should receive a comprehensive exit ‘care pack’, containing individualized key learnings, reflections, resources, tools, and referrals or contact details for support. Young people and carers suggested this could also include gifts to help maintain progress after leaving, based on items they had used during their stay, such as fidgets or art/crafts. The idea, drawn from Maytree, of providing a personal letter upon exit that celebrates the young person’s progress was well received across groups, though participants emphasized it should be genuine, heartfelt, personalized and considerate of varying literacy levels.

Views on involvement in the exit process showed both shared perspectives and differences. Across groups, there was recognition that stakeholders and/or carers could be an important source of support when a young person returns home. Stakeholders and carers wanted specific information on how they can assist after exiting, but young people emphasized retaining control over who was included in their exit plan. Stakeholders suggested including a safety plan as part of the exiting process, whereas young people felt safety plans need to be individualized to be helpful. An alternative suggestion was to create an action plan with the young person that included both short-term goals (e.g., arranging a coffee catch up with a friend) and long-term goals (e.g., apply for study).

#### 3.5.3. Follow-Up

Participants unanimously agreed that follow-up contact with the young person after exiting Luminos was essential. However, opinions differed on timing between the groups. Young people suggested initial follow-up times ranging between two days to one month, carers suggested between one day to two weeks, and stakeholders recommended follow-up one month after leaving Luminos. Similar variation emerged regarding how long follow-up contacts should continue. Young people proposed timeframes ranging from seven days to six months, carers between three and six weeks, and stakeholders over a few months.

Across groups, participants agreed that young people’s preferred mode of communication should be respected. Young people preferred receiving a text message before any phone calls and requested that the same staff members who had supported them at Luminos manage their follow-ups. Participants also agreed that follow-up contacts should be meaningful, such as discussing their progress towards achieving goals from their action plan and referrals to services if additional support is needed. Carers, however, highlighted the need to consider staff workload so that follow-ups do not become overly burdensome.

#### 3.5.4. Re-Using the Service

All groups agreed that Luminos should not operate as a single-entry service model, as young people’s needs can change over time. Young people indicated that a single-entry model could be anxiety provoking, creating pressure to attend at the “perfect” and potentially delaying seeking help until their situation worsens. However, participants recognized that allowing re-entry could increase wait times and risk service dependence. A proposed compromise was to limit re-entry to once every 6 to 12 months.

There was also agreement across groups that an ‘outpatient’ service could be beneficial, enabling young people to maintain connections with staff they had engaged with during their stay. Suggested options for non-residential involvement included therapeutic workshops, volunteering and casual social interactions.

#### 3.5.5. Linking to Other Services

Providing tailored referrals was seen as a core responsibility of Luminos, with agreement across groups that this should include warm referrals, or as a minimum, assisting the young person connect with a service. Young people highlighted clinical services, such as counselling, group therapy and headspace centers (publicly primary care mental health services in Australia) as suitable referral options. Young people and stakeholders emphasized the importance of referrals to services addressing social determinants of health, including physical health, housing, employment, domestic and family violence, legal, education, training, financial assistance, Centrelink (the Australian benefits office) and foodbanks. They also valued help in linking with community-based social activities, such as hobby activities and social groups.

### 3.6. Challenges and Safety

#### 3.6.1. Operational Challenges

The potential for waitlists was identified as a significant challenge, with some stakeholders particularly concerned about the impact on Aboriginal young people, who already face barriers and delays in service access. While acknowledging this issue, participants stressed the importance for Luminos to provide interim support for those waiting, including referrals to appropriate services and regular check-ins. Ensuring the security and privacy of Luminos was another priority, suggestions included secured entry into the house and keeping the service location undisclosed. Stakeholders also raised concerns that securing sustainable funding for Luminos would be an ongoing challenge.

#### 3.6.2. Interactions with Others

Interpersonal connection among guests were seen as valuable for enjoyment, distraction and healing across all groups, making the chance to form friendships an important aspect of the stay. At the same time, participants raised concerns about potential issues arising from interactions between guests, which included the possibility of intimate or sexual relationships, and a broad age range. Young people and carers suggested staggered intakes by age as a possible solution.

“*If you’ve got someone who is, let’s say, depressed and anxious, and then you’ve got someone who is really out there, who is a real extrovert… How do you set the culture of the sanctuary? How do you work out what that culture is and keep it? Because the dynamic is absolutely contagious. And if you’ve got two people who have a history of eating disorders, how do you make sure that can’t flourish in this environment? And what do you allow? Can they go into each other’s rooms?*”(Carer 1)

“*Sometimes the group dynamics can be a little bit complicated… I know of kids who meet each other and then go on to form friendships that are potentially not the healthiest connections. I guess it would be a bit of a concern that there is scope for that to happen again*.”(Stakeholder 14)

Concerns about disrespectful or discriminatory behavior between guests, particularly towards LGBTQA+ people and the potential for homophobia, were voiced by young people and carers. They also expressed concerns about the risk of exposure to traumatic experiences, such as witnessing distress or hearing self-disclosures of suicidal experiences. Carers shared examples of how their child had been impacted by similar issues in other residential facilities, and young people requested that clear boundaries be established for guest interactions.

For some young people, interacting with peers in the house may be anxiety provoking, driven by fears of rejection or isolation. Worries about confidentiality breaches, either during the stay or through encounters with fellow guests or staff outside of their stay, were also reported. Participants also voiced concerns about the possibility of harmful behaviors developing or intensifying in the house, such as competitive behavior related to eating disorders or the uptake of smoking.

#### 3.6.3. Eligibility for Service

Participants highlighted challenges with the eligibility criteria, as experiences of suicidality often fluctuate. Young people felt that eligibility needed to be clearly communicated, as “passive” suicidal ideation could have different connotations. Stakeholders expressed concern that young people might feel compelled to conceal the true extent of their suicidality to gain acceptance into Luminos.

#### 3.6.4. What Would Prevent People from Using Luminos?

The primary barrier for young people in entering Luminos was if it resembled a clinical setting, with general aesthetics and operational components (i.e., bringing personal items, leave policy) also raised. Young people were also concerned about being separated from their pets and perceived family judgement. Carers noted that exclusion from their young person’s care whilst at Luminos would be a barrier.

### 3.7. Measuring Outcomes

#### 3.7.1. Measures of Success

The most significant measure of success was young people leaving with a “hope for the future.” Additional indicators of success included a more positive outlook on life, clearer future aspirations, and a stronger sense of purpose. Participants also believed that a successful outcome would involve young people leaving with a ‘psychological toolkit’ that includes self-regulation, help-seeking strategies, improved anxiety management, and awareness of available services.

Young people wanted to leave Luminos feeling stronger and more empowered. They expressed a desire to experience an improved quality of life with reduced suicidal feelings. They hoped to have the opportunity to process past experiences and gain a better understanding of their own mental health. Carers hoped that young people would leave Luminos with increased resilience, self-worth, confidence, self-advocacy, and personal agency. They desired for young people to feel more ‘mentally sound’ and motivated to participate in life activities. They also wanted young people to develop a stronger sense of identity beyond their mental health and experience reduced shame about it. Stakeholders echoed this, wanting young people to feel more ‘mentally realigned’ and experience improvements in self-worth, self-esteem, confidence, self-awareness, autonomy, and control.

“*With my child, we talk about bees in a bonnet. At some point, she’s got too many bees and she has to get rid of them. So, I think my expectation is that some of the bees just disappear*.”(Carer 7)

“*Not to be consumed by your mental health*.” (Young person 33). Followed by: “*Exactly, just be a normal person*.”(Young person 35)

“*I would like you to come away with a sense that, ‘hey, look, things are pretty crap at the moment, but it’s not always going to be like this, and there’s other people that feel like this as well*’”(Carer 8)

#### 3.7.2. Data Collection Methods

Participants agreed that collecting data, using both quantitative and qualitative methods, was essential for assessing the impact of Luminos. Questionnaires were deemed appropriate but various formats were suggested (e.g., paper-based, online, or verbally), including distributing links via email, QR codes, or in-house iPads. Participants stressed surveys should be short. Some young people suggested tracking their mood over their stay through visual mood charts, diaries or app-based surveys, however others felt this would be overwhelming and intrusive. Participants also suggested gathering feedback from carers and stakeholders who engage with Luminos.

There was discrepancy among participant groups on the timing of data collection points. While there was consensus that questions should be asked at the beginning, after the stay and in a follow-up, the specific timing for each stage varied. Participants noted that young people should have the option to answer questions at their own pace and not be required to do so immediately upon arrival. Young people indicated that they might not be in the right mindset to answer questions upon exiting, and doing so could negatively affect their departure. Stakeholders noted the risk of disengagement when attempting to collect post-stay data but felt follow-up questionnaires should be completed within 1–2 months after leaving Luminos.

Participants were asked whether it would be acceptable to anonymously track young people’s healthcare data, such as visits to emergency departments, GP, and psychology appointments (referring to the use of linked administrative data). Most young people were accepting of this, particularly because it would reduce the need for them to answer questions, but emphasized the importance of voluntary informed consent and a clear explanation of how the data would be used and for what purpose. However, carers and stakeholders did not support tracking healthcare data due to ethical concerns about overall intrusiveness. They also raised questions about the accuracy of the data, noting that it might not sufficiently differentiate between physical and mental health, and debated whether accessing support reflects poorer health or improved help-seeking behaviors. Details of the questions suggested to measure outcomes can be seen in Table 3.

## 4. Discussion

Currently in Australia and globally, the medical framework predominates service responses to suicide intervention, with EDs as the primary point of contact for youth in suicidal crises. Whilst the medical framework approach to suicide intervention provides essential crisis intervention and some forms of treatment, it often falls short in addressing the complex, multifaceted needs of suicidal youth, which require a more holistic and youth-led approach. The current findings provide a critical lens on how relational, non-clinical approaches to suicidal distress can be adapted for young people in developmentally and culturally responsive ways. While existing models such as Maytree (UK) [9] and SPARC (Australia) [10] offer important precedents, they were designed for adults. Luminos builds on their foundational principles—non-medical support, emotional respite, and peer-led connection—but tailors them to the distinct needs, preferences, and developmental contexts of young people.

A key innovation lies in how relational care was operationalized for youth. While Maytree focusses on befriending through emotional openness and intimacy, encouraging adults to share deeply in a safe, non-judgmental space, Luminos participants valued emotional connection but emphasized the importance of choice—being able to control when and how to engage emotionally—and cultural safety, which includes recognition and respect for diverse identities. Young people preferred relationships that affirmed their developing sense of self and autonomy rather than those focused primarily on emotional disclosure. Peer staff were valued not only for shared lived experience but also for their ability to relate authentically without hierarchical or clinical authority. This non-hierarchical approach reduces power imbalances, which young people identified as barriers to trust and openness. Thus, while Maytree’s relational care emphasizes emotional closeness as a path to healing [9], Luminos adapts these principles to foreground empowerment, respect for identity, and voluntary engagement—priorities that better reflect youth developmental and cultural contexts [14,15].

Duration and repeat utilization of care highlights the unique challenges young people in suicidal distress face in having their complex and evolving needs met within fixed, short-term program durations. While five days reflects the length of established models like Maytree, many participants—particularly Aboriginal young people—expressed the need for longer stays to build trust, establish therapeutic relationships, and ensure culturally safe support. Extended and flexible timeframes provide young people the necessary space to feel heard and understood—foundational for effective intervention—while better aligning with the fluctuating nature of suicidal distress and when they are ready to engage, thereby enhancing the service’s impact.

The single-entry model highlights a key distinction between youth and adult needs: while adult services often operate under the assumption that a single episode of care is sufficient, young people often feel they need opportunities for re-engagement due to the fluctuating nature of their emotional states and developmental processes. Young people described how a one-off opportunity to access the service could heighten anxiety and delay engagement until distress had intensified, ironically working against the early intervention ethos the service aims to uphold. The unpredictability of emotional distress and the importance of cultural safety mean that brief, one-time stays may not offer sufficient time or opportunity for meaningful engagement. This underscores the necessity of viewing Luminos not as an isolated intervention but as part of a broader system that supports young people through multiple points of contact and sustained relational care.

Luminos shares key structural features with existing residential suicide respite services such as Maytree and SPARC, including short-term stays with a combination of shared meals, structured group activities, and personal downtime. These elements create a balanced daily structure that supports both social connection and individual recovery. This program structure reflects a core principle of providing consistent, supportive routines in a non-clinical environment. By embedding flexibility into a structured environment, Luminos addresses potential tensions between providing safety through routine and respecting young people’s evolving needs for control and identity affirmation.

The co-design process surfaced several unresolved tensions that have important implications for the operation of Luminos and similar youth-led respite models. These are not minor implementation hurdles, but meaningful challenges that reflect the complexity of delivering safe, inclusive, and developmentally appropriate care in a residential setting.

One recurring tension involved the regulation of phone use. While young people viewed mobile phones as critical for maintaining a sense of connection and autonomy, carers and some stakeholders raised concerns about the potential for triggering content, disrupted sleep, or privacy violations. This conflict reveals a critical challenge for youth suicide respite services: how to uphold young people’s agency and relational needs without compromising safety or therapeutic focus. Addressing this requires tailored, context-sensitive approaches rather than rigid, one-size-fits-all rules.

Peer interactions within residential care represent one of the most nuanced and sensitive aspects of youth suicide respite [16,17,18]. While connecting with peers who share similar experiences can provide validation and reduce feelings of isolation, these interactions also carry risks unique to young people, such as re-traumatization, breaches of confidentiality, stigma, and the blurring of professional boundaries. This complexity underscores that peer support in youth settings is not inherently therapeutic; rather, it requires thoughtful and proactive management tailored to the developmental and emotional needs of young people [19]. This highlights the essential need for youth-focused services to carefully consider and actively manage peer dynamics to create a safe and supportive environment.

The challenges inherent in managing peer interactions are compounded by concerns related to sexuality, gender identity, and trauma, demanding careful consideration to maintain safety and inclusivity within residential care environments. Concerns about homophobia and discriminatory behavior underscore the need for consideration of LGBTQA+ young people in mixed-gender or non-specific intake environments, indicating that without targeted safeguards, these spaces may unintentionally replicate harmful social dynamics. Anxiety about exposure to others’ distress, often rooted in previous negative experiences in hospital settings [2,18,20], underscores the need to carefully balance fostering peer connection with protecting against re-traumatization and emotional overload. These considerations emphasize the necessity of moving beyond standardized protocols to personalized, trauma-informed approaches that anticipate and address risks related to identity-based vulnerabilities and emotional contagion in shared living spaces. Such complexities reinforce the importance of flexible intake processes, in-depth pre-admission conversations, and careful guest matching that prioritizes psychological and cultural safety over administrative convenience.

This co-design highlights specific service elements that should be prioritized when developing youth-specific suicide respite models, recognizing their needs differ substantially from adults. These include embedding a relational approach that centers trust and emotional connection, ensuring flexible opportunities for re-engagement, and creating environments that are non-clinical and youth oriented. While local co-design remains essential to tailoring services to specific communities [21,22,23], these core elements offer a practical foundation for informing the development of other youth-focused models. While rich in depth and grounded in co-design principles, this evaluation was based on data from a single site and may not capture challenges, perspectives or operational changes that will likely emerge as the service is implemented and used more widely or over a longer period. Additionally, participant recruitment relied on voluntary engagement, and although the young cohort reflected diverse identities and experiences, the findings could represent those more inclined toward non-clinical or peer-based models of support. Constraints emerged during the design process where participant suggestions could not be fully realized due to practical or financial limitations or reflected misunderstandings about Luminos’ intended function. For example, proposals such as a basketball ring or sensory room exceeded the spatial capacity of the residential property in inner Perth, while expectations for extensive staffing (e.g., five staff during the day and similar numbers overnight) were not feasible within the funding parameters of a no-cost service. There was also occasional confusion regarding Luminos’ role, with it being misinterpreted as a crisis intervention service rather than a non-clinical respite model.

Future steps for Luminos involve evaluating its effectiveness over an extended period of operation. If shown to be beneficial for young people experiencing suicidal distress, the model may offer potential for expansion into other locations or adaptation for specific marginalized groups. Adapting the model for different contexts, such as rural or Aboriginal and Torres Strait Islander communities, requires careful consideration of a range of structural and contextual factors. These may include geographic isolation, limited access to youth-specific or culturally safe services, and challenges in recruiting and retaining appropriate staff. Another important consideration is whether the host organization is willing and able to hold high levels of suicide risk, including in minors. Additionally, the need to embed local cultural practices and community priorities into the design is critical. Addressing these issues goes beyond tailoring surface features of the model; it requires meaningful partnerships, sustained community engagement, and policy and funding frameworks that support flexible, place-based implementation.

Financing a residential suicide respite service involves substantial costs. However, these expenses may be offset by potential cost savings if the service effectively reduces the need for ED visits and inpatient psychiatric admissions among young people [24]. By providing timely, developmentally appropriate, and culturally safe support, such services have the potential to not only improve outcomes for youth but also alleviate pressure on more intensive and costly parts of the healthcare system, thereby representing a cost-effective investment in youth mental health care.

## 5. Conclusions

The co-design of Luminos highlighted a decisive shift away from conventional models of responding to youth suicidal distress, centering relational care, cultural safety, and youth autonomy within a flexible, homely environment. Three priorities emerged: (1) the need for spaces that feel non-clinical and allow privacy and personalization; (2) the importance of relational, consistent, and diverse staff, including peers with lived experience; (3) the value of flexible structures that adapt to individual needs, triggers, and cultural contexts. These findings have direct implications for both Luminos and the wider system: environments must be deliberately designed to feel safe, comfortable, and affirming; staffing must encompass lived experience and cultural diversity to build trust; and operational frameworks should remain flexible enough to respond to fluctuating needs while maintaining a sense of safety. As health systems seek more responsive and humane alternatives to emergency-based care, Luminos offers a promising youth-led model—one that challenges assumptions about risk, safety, and support in the context of suicidality.

## Figures and Tables

**Figure 1 ijerph-22-01449-f001:**
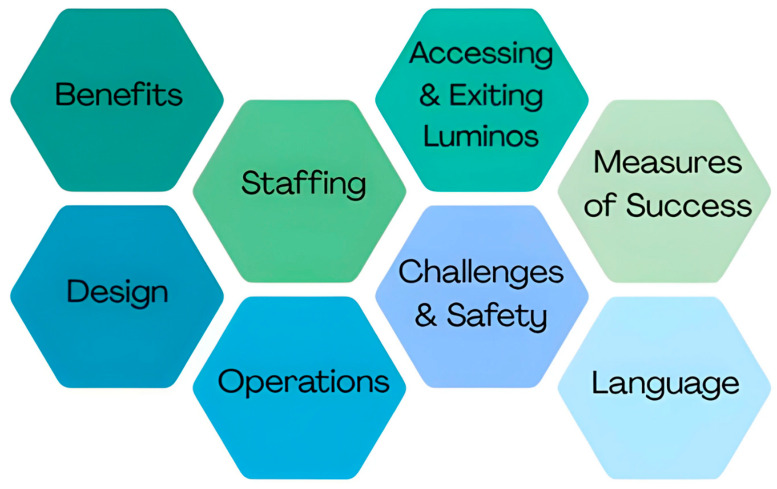
Themes explored.

**Figure 2 ijerph-22-01449-f002:**
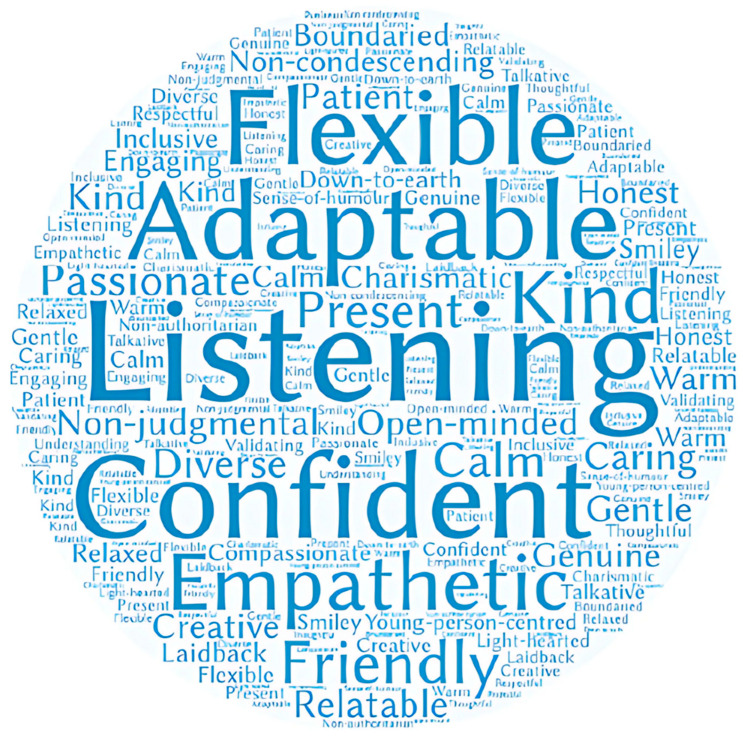
Word cloud of staff qualities.

**Table 1 ijerph-22-01449-t001:** Demographic Information for Young People *.

Characteristic		
Age		*M* = 20.5 years (*SD* = 2.5) *Range 16*–*25*
		*N*
Gender Identity	Female Male Non-binary Transmasculine Gender-fluid	19 8 7 2 1
Identity	Aboriginal and/or Torres Strait Islander	3
	LGBTQA+	20
	Neurodivergent	16
	Migrant or Refugee Background	8
	Disability	13
Care and Justice Experience	Experience in Out-Of-Home Care	5
	Involvement with Justice System	6
Living Circumstances	Parents On Own Partner Friends Supported Accommodation Other Family	15 7 4 3 3 3

*** Some participants chose not to provide this information.

**Table 2 ijerph-22-01449-t002:** Activities.

Category	
Structured/Organized Activities	Group excursions (e.g., visiting cafes, galleries, shopping, picnics) Exercise (e.g., walks, yoga, dancing, soccer, basketball) Cooking and baking Gardening Creative activities (e.g., painting, drawing, creative writing, knitting, jewelry making)
Unstructured Activities	Exercise (e.g., provision of boxing bag, basketball hoop, yoga mats) Reading Music instruments Art and craft (e.g., coloring books and paints) Journalling Card and board games Jigsaw puzzles Technology-based (e.g., watching TV/movie, gaming)
Therapeutic Activities	One-on-one talking Group reflections Mindfulness Meditation Art therapy Music therapy Animal therapy Distress tolerance
Training Activities	Psychoeducation (e.g., managing emotions, building resilience, goal planning and problem-solving skills) Self-care (e.g., healthy habits and sleep hygiene) Life skills (e.g., budgeting, cooking, writing a resume, car maintenance, first aid) Harm reduction strategies Self-advocacy

**Table 3 ijerph-22-01449-t003:** Suggested questions for evaluation.

	Young People	Carers	Stakeholders
**Satisfaction**	Did you enjoy your stay? What did you enjoy about the sanctuary? How satisfied were you with your stay? What workshop or activity stood out to you and why?	How do you feel about your stay?	What did you like or dislike? What was your favorite part?
	Would you recommend this service to friends?		Would you recommend this service to friends?
	Did you enjoy the structure and build of the building and outdoor area? Did you enjoy the structure of the activities and the independence you had?		Would you come back again?
	Did you feel this was relevant to you?		Do you feel your culture and identity were respected and valued?
**Changes to service**	What would you change? What do you think can be improved? What services do you think need to be added?		How can we improve this service? Is there anything that you’d change? Is there anything that can be done better?
**Skills learnt**	What have you learnt? Has the stay benefitted your child? What did you find helpful?	Current coping strategies	What have you learnt?
**Connections**	Did you enjoy the company of other young people/other workers—why/why not?		Is there someone you connected with in this program?
**Improved MH/wellbeing**	How are you feeling? Validated mental health scales, e.g., depression, anxiety, suicide		What are your energy levels like? Recency/frequency/intensity of suicidal thoughts
**Other outcomes**		Where do you see yourself in a month? What are you looking forward to? What do you like doing? What interests you? What drives you? What’s your passion?	
**Long term**	Have you found any areas improved? Has anything come up that you learnt how to navigate better while in the service? How did you find your transition back into your home environment?	What did you do today (e.g., did you leave the house)?	Do you feel like the supports and skills provided to you are going to benefit your recovery in the long run? Do you feel like your mental health is going to improve if you apply these skills?
**Safety**	Did you feel hurt? How safe did you feel?		

## Data Availability

The original contributions presented in this study are included in the article. Further inquiries can be directed to the corresponding author.

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
