# Peer review of "The Luminos Project: Co-Designing a Short-Stay Suicide Support Model for Young People"

_ijerph, 2025, doi:10.3390/ijerph22091449_

Round 1
Reviewer 1 Report
Comments and Suggestions for Authors
This study conducted a thematic analysis of key elements that constitute the Luminous Project, a short-stay non-medical residential intervention program for adolescents with suicide ideation. Based on interviews and focus group discussions, the study identified eight core themes using thematic analysis. The research topic is important, and identifying the core themes that constitute the Luminous project would provide valuable insights and transferrable models for suicide intervention in other countries and regions. That said, the manuscript itself needs some reorganization to clarify the research questions and strengthen its overall presentation.
- The Introduction section mentioned the downsides of traditional suicide intervention approaches, and the Maytree model, as well as the Luminous Project, were hailed as promising alternatives. But here these alternatives were only briefly introduced. Considering that the Journal is targeted at international readership, it would be preferable that the details of the two models could be elaborated. Some statistics of both programs, if available, would help illustrate their effectiveness.
- The research question of this study is not clearly stated at the end of the Introduction. Is it a study aimed at “uncovering/describing” themes of a successful project, “identifying” potential rooms for improvement, or a combination of the two?
- Details of the interview and focus group are lacking. What kind of questions were asked?
- The discussion section is underdeveloped. The findings should be integrated with theoretical perspectives, and this could enhance the interpretative depth and theoretical contribution of the study.
- Implications and limitations are completely absent.
Author Response
1.
This study conducted a thematic analysis of key elements that constitute the Luminous Project, a short-stay non-medical residential intervention program for adolescents with suicide ideation. Based on interviews and focus group discussions, the study identified eight core themes using thematic analysis. The research topic is important, and identifying the core themes that constitute the Luminous project would provide valuable insights and transferrable models for suicide intervention in other countries and regions. That said, the manuscript itself needs some reorganization to clarify the research questions and strengthen its overall presentation.
Response: Thank you. We believe the manuscript is much improved after making substantial revisions.
2.
The Introduction section mentioned the downsides of traditional suicide intervention approaches, and the Maytree model, as well as the Luminous Project, were hailed as promising alternatives. But here these alternatives were only briefly introduced. Considering that the Journal is targeted at international readership, it would be preferable that the details of the two models could be elaborated. Some statistics of both programs, if available, would help illustrate their effectiveness.
Response: We have extended the introduction and added more information about the models. We have included more information about the evaluation findings of Maytree. We have also discussed Australia’s Suicide Prevention and Recovery Centre (SPARC), a peer-support residential stay, as another alternative.
3.
The research question of this study is not clearly stated at the end of the Introduction. Is it a study aimed at “uncovering/describing” themes of a successful project, “identifying” potential rooms for improvement, or a combination of the two?
Response: We have rewritten a clearer aim of the study to state:
“This study aims to adapt and incorporate foundational principles—such as peer support, emotional connection, and trauma-informed care—into a model designed specifically for young people in Western Australia. Through co-design consultations with young people, caregivers, and stakeholders in Perth, it
identifies core elements that define an effective youth-centered, trauma-informed suicide respite service. The findings provide an evidence-based framework to guide the creation of a feasible, responsive alternative to conventional clinical interventions, supporting more meaningful and accessible suicide support for young people.”
4.
Details of the interview and focus group are lacking. What kind of questions were asked?
Response: We have added additional information in the Procedures section:
“Separate interview guides were developed for each participant group, tailored to their specific experiences and insights. Group discussions were guided by a series of exploratory questions designed to inform the co-design of the youth suicide service model. These questions explored participants’ perspectives on service design, key features of care (including trauma-informed principles), roles and responsibilities of staff and volunteers, training and workforce requirements, referral and access pathways, length of stay and transitions, involvement of families and carers, and anticipated implementation challenges.”
5.
The discussion section is underdeveloped. The findings should be integrated with theoretical perspectives, and this could enhance the interpretative depth and theoretical contribution of the study.
Response: We have changed the discussion considerably and developed a better discussion of the contribution of the study. We hope the reviewer feels that their concerns were addresses.
6.
Implications and limitations are completely absent.
Response: We have included sections in the revised Discussion that examine the implications and limitations of this work.
Reviewer 2 Report
Comments and Suggestions for Authors
Dear authors, thank you for the opportunity to review this important manuscript. The study addresses a critical public health issue in Australia by presenting short-stay suicide support service tailored to the needs of young people. However, to enhance the clarity, transparency, and broader applicability of the findings and increase scientific impact of your paper, I offer the some comments and suggestions for minor revision.
-
I suggest quantifying suicide in the first sentence of the abstract. This would emphasize the importance of the issue and provide the reader with a clearer understanding of its scale in Australia.
-
A comparative trend could also be included (e.g., "an X% increase over the past decade") to highlight the dynamics of the problem.
-
The transition from the problem statement to the Maytree model could be made more logically coherent.
-
No demographic information is provided (e.g., gender, ethnicity, LGBTQ+ status, previous system involvement).
-
There is also no mention of whether written informed consent was obtained from participants. Please specify the exact type of consent used.
-
Additionally, were young participants aged 16–17 required to obtain parental consent, or were they permitted to consent independently?
-
There is no explanation of who conducted the data analysis — how many researchers were involved, and whether inter-coder reliability was assessed?
-
All tables should be formatted according to MDPI requirements.
-
The source of each quotation should be clearly indicated (e.g., YP01, ST03) to assess representativeness.
-
A key recommendation is to summarize the main conclusions more clearly and to highlight differences across the participant groups.
-
It would also be valuable to mention which themes the researchers anticipated but did not emerge — possibly due to their sensitivity or social taboo.
-
The discussion remains too general. It does not compare specific aspects of the Luminos model with other existing alternatives worldwide (e.g., crisis hostels, safe haven cafés, peer respite centers).
-
I suggest including examples of evaluations of similar services (including Maytree, if available), and positioning Luminos within a broader international context.
-
Is there any reference to Luminos' real-world implementation to date, such as the number of people accommodated, age breakdowns, or usage patterns?
-
The discussion does not elaborate on some of the most interesting and delicate findings from the results:
– Disagreements about phone use
– Parental resistance and the need for control
– The dilemma regarding the short 5-day stay
– Possibility of harmful peer interactions
– Questions surrounding sexuality and potential retraumatization -
These findings deserve to be discussed explicitly.
-
The limitations of the study should also be more clearly acknowledged.
-
For example, young people often describe ideal or desirable service features, not necessarily operationally feasible ones.
-
It would be useful to reflect on what structural and contextual factors are needed for scaling the model (e.g., budget, legislation, community support), and how it could be adapted for rural, Indigenous, or culturally diverse contexts.
-
Consider shortening the title — for example:
“The Luminos Project: Co-Designing a Short-Stay Suicide Support Model for Young People.” -
The phrase “we hope this co-design will inform similar services” is vague and passive. A more effective formulation might be:
“These findings provide actionable insights for the development of alternative, youth-informed suicide support services.” -
Finally, the abstract could benefit from a slightly more academic tone — with fewer vague phrases and a clearer emphasis on the importance of both the issue and the findings.
Author Response
1.
Thank you for the opportunity to review this important manuscript. The study addresses a critical public health issue in Australia by presenting short-stay suicide support service tailored to the needs of young people. However, to enhance the clarity, transparency, and broader applicability of the findings and increase scientific impact of your paper, I offer some comments and suggestions for minor revision.
Response: Thank you. The paper has been much improved by the reviewer feedback.
2.
I suggest quantifying suicide in the first sentence of the abstract. This would emphasize the importance of the issue and provide the reader with a clearer understanding of its scale in Australia.
Response: We have changed the opening sentence in the abstract to: “Suicide was the leading cause of death among young Australians aged 15-24 years old, with 298 lives lost in 2023.”
3.
A comparative trend could also be included (e.g., "an X% increase over the past decade") to highlight the dynamics of the problem.
Response: We have attempted to address this by including a comparative trend statement in opening paragraph of the Introduction section.
“Over the past decade, there has been a significant and concerning increase in youth suicide rates, reaching 31.8% among 15–17-year-olds and 33.1% among 18–24-year-olds in 2023. These figures were notably lower in 2021- at 16.5% and 23.9%, respectively- underscoring the urgent need for innovative and effective approaches to suicide prevention and support [1]”
4.
The transition from the problem statement to the Maytree model could be made more logically coherent.
Response: We have re-worded the paragraph to introduce relational approaches as one alternative to suicide intervention and then introduce Maytree as an example of relational approaches.
5.
No demographic information is provided (e.g., gender, ethnicity, LGBTQ+ status, previous system involvement).
Response: A table presenting demographic information of young people who participated has now been included (see Table 1).
6.
There is also no mention of whether written informed consent was obtained from participants. Please specify the exact type of consent used.
Response: The following has been added to the Participants section:
“All participants provided written informed consent, either electronically via Qualtrics or in person. Young people aged 16-17 years were consented as mature
minors. Participants under the age of 16 required written consent from a parent or guardian.”
7.
Additionally, were young participants aged 16–17 required to obtain parental consent, or were they permitted to consent independently?
Response: This has now been addressed (see point 6).
8.
There is no explanation of who conducted the data analysis — how many researchers were involved, and whether inter-coder reliability was assessed?
Response: There are many methods that can be used to ensure rigour in qualitative health research. Double coding, as suggested by examiner one, is another method that can be used, however, it is also contentious within the qualitative methodology literature (see O’Connor, C., & Joffe, H. (2021). Intercoder reliability in qualitative research: Debates and practical guidelines. International Journal of Qualitative Methods, 19. https://doi.org/10.1177/1609406919899220)
Furthermore, double coding does not align theoretically with the constructivist epistemology that guided the research or Braun & Clarke’s thematic analysis approach that was used to analyse the data (see Braun, V., & Clarke, V. (2021). One size fits all? What counts as quality practice in (reflexive) thematic analysis? Qualitative Research in Psychology, 18:3, 328-352, https://doi.org/10.1080/14780887.2020.1769238)
We have added substantial detail the Analysis section:
“Thematic analysis was conducted following Braun and Clarke’s [12] six-phase framework: (1) data familiarisation, (2) generation of initial codes, (3) identification of potential themes, (4) theme review, (5) defining and naming themes, and (6) report production. Author LH began by engaging thoroughly with the data through repeated transcript readings to ensure comprehensive familiarity. Coding was undertaken in NVivo using a hybrid deductive–inductive approach. Deductive codes were derived from the pre-established categories embedded within the interview and focus group guides, providing an initial organisational framework. Concurrently, inductive codes capturing salient, emergent concepts within the data were developed. Initial coding and theme development were conducted by LH and subsequently reviewed and refined by SS, who had conducted the majority of interviews and focus groups and reviewed all transcripts for accuracy. Inductive codes were grouped into sub-themes, which were iteratively reviewed and integrated with the deductive framework to form a cohesive thematic structure that balanced participant-driven insights with the study’s predetermined focus areas.
Interviews were conducted with participants from each participant group until thematic saturation was reached. While concept of saturation is debated in qualitative research [13, 14], it is generally understood as the point at which
additional data no longer yield novel insights or themes. In this study, saturation was reached when no further themes arose within each participant group. Ongoing, iterative analysis conducted alongside data collection allowed researchers to monitor the recurrence of ideas in line with the study’s aim.”
9.
All tables should be formatted according to MDPI requirements.
Response: This has been amending according to the MDPI information we are able to find.
10.
The source of each quotation should be clearly indicated (e.g., YP01, ST03) to assess representativeness.
Response: The source of each quotation has now been included to enhance transparency and allow assessment of representativeness across participant groups.
11.
A key recommendation is to summarize the main conclusions more clearly and to highlight differences across the participant groups.
Response: We have attempted to do this throughout the Results section. It has been revised substantially.
12.
It would also be valuable to mention which themes the researchers anticipated but did not emerge — possibly due to their sensitivity or social taboo.
Response: There is nothing specific that we thought would be mentioned by participants that they didn’t mention. We feel that participants were very open and honest and frank about their needs.
13.
The discussion remains too general. It does not compare specific aspects of the Luminos model with other existing alternatives worldwide (e.g., crisis hostels, safe haven cafés, peer respite centers).
Response: We have changed the Discussion considerably and we hope these concerns are now addressed.
14.
I suggest including examples of evaluations of similar services (including Maytree, if available), and positioning Luminos within a broader international context.
Response: We have attempted to address this comment in the Introduction and Discussion sections of the paper, both of which have been substantially revised.
15.
Is there any reference to Luminos' real-world implementation to date, such as the number of people accommodated, age breakdowns, or usage patterns?
Response: As the evaluation is still in progress, it is premature to report on service usage or demographic data at this stage. This paper is focused specifically on the co-design process. Findings related to real-world implementation, including accommodation numbers, age breakdowns, and usage patterns, will be presented in a separate paper upon completion of data collection. We also have a paper planned on the model of care. There is simply too much information for one manuscript.
16.
The discussion does not elaborate on some of the most interesting and delicate findings from the results: – Disagreements about phone use – Parental resistance and the need for control – The dilemma regarding the short 5-day stay – Possibility of harmful peer interactions – Questions surrounding sexuality and potential retraumatization
These findings deserve to be discussed explicitly.
Response: We have attempted to elaborate on these findings in the Discussion section as suggested.
17.
The limitations of the study should also be more clearly acknowledged. For example, young people often describe ideal or desirable service features, not necessarily operationally feasible ones.
Response: We have now included paragraphs discussing implications and limitations in the revised Discussion section.
18.
It would be useful to reflect on what structural and contextual factors are needed for scaling the model (e.g., budget, legislation, community support), and how it could be adapted for rural, Indigenous, or culturally diverse contexts.
Response: We have attempted to describe these factors in the revised Discussion.
19.
Consider shortening the title — for example: “The Luminos Project: Co-Designing a Short-Stay Suicide Support Model for Young People.”
Response: The title has been shortened as suggested.
20.
The phrase “we hope this co-design will inform similar services” is vague and passive. A more effective formulation might be: “These findings provide actionable insights for the development of alternative, youth-informed suicide support services.”
Response: We have changed that sentence in the abstract to: “These findings provide actionable insights for the development of alternative, youth-informed suicide support services.”
We have changed the last section in Introduction to: “Through co-design consultations with young people, caregivers, and stakeholders in Perth, it identifies core elements that define an effective youth-centered, trauma-informed suicide respite service. The findings provide an evidence-based framework to guide the creation of a feasible, responsive alternative to conventional clinical interventions, supporting more meaningful and accessible suicide support for young people.”
21.
Finally, the abstract could benefit from a slightly more academic tone — with fewer vague phrases and a clearer emphasis on the importance of both the issue and the findings.
Response: We have rewritten the abstract in line with this suggestion.
Reviewer 3 Report
Comments and Suggestions for Authors
I have reviewed the manuscript titled “Co-Designing a Short-Stay Model of Care for Young People with Lived Experience of Suicidal Thoughts and Behaviours: The Luminos Project.” This study presents a timely and socially significant initiative grounded in lived experience to develop a non-clinical, short-stay intervention model for suicidal youth in Western Australia. From a psychiatric perspective, the manuscript demonstrates strong conceptual grounding and commendable ethical sensitivity, yet it requires major revision to meet methodological and reporting standards expected in peer-reviewed psychiatric and public health journals. The primary limitation lies in the absence of a clear methodological justification for the hybrid thematic analysis and lack of transparency in coding procedures. There is no discussion on intercoder reliability or data saturation, which are essential to assess thematic robustness. Additionally, while the co-design approach is well-articulated, the study lacks an ablation or comparative analysis—e.g., comparing participant groups or evaluating different versions of the model through iteration—to substantiate the service design's efficacy. The manuscript would benefit from a clearer articulation of exclusion/inclusion criteria for participant recruitment and more explicit detailing of how cultural safety principles were integrated beyond listing stakeholder categories. Importantly, while the service has been implemented, no preliminary evaluation results or implementation fidelity data are provided, limiting the ability to assess potential clinical impact. A future direction and limitations section should be added to critically reflect on scalability, sustainability, and potential risks of peer-led models in high-risk populations.
Comments on the Quality of English LanguageThe English could be improved to more clearly express the research.
Author Response
22.
I have reviewed the manuscript titled “Co-Designing a Short-Stay Model of Care for Young People with Lived Experience of Suicidal Thoughts and Behaviours: The Luminos Project.” This study presents a timely and socially significant initiative grounded in lived experience to develop a non-clinical, short-stay intervention model for suicidal youth in Western Australia. From a psychiatric perspective, the manuscript demonstrates strong conceptual grounding and commendable ethical sensitivity, yet it requires major revision to meet methodological and reporting standards expected in peer-reviewed psychiatric and public health journals.
Response: Thank you for the feedback. We hope that we have addressed the concerns of the reviewers.
23.
The primary limitation lies in the absence of a clear methodological justification for the hybrid thematic analysis and lack of transparency in coding procedures.
Response: We have added substantial detail the Analysis section:
“Thematic analysis was conducted following Braun and Clarke’s six-phase framework [12]: (1) data familiarisation, (2) generation of initial codes, (3) identification of potential themes, (4) theme review, (5) defining and naming
themes, and (6) report production. Author LH began by engaging thoroughly with the data through repeated transcript readings to ensure comprehensive familiarity. Coding was undertaken in NVivo using a hybrid deductive–inductive approach. Deductive codes were derived from the pre-established categories embedded within the interview and focus group guides, providing an initial organisational framework. Concurrently, inductive codes capturing salient, emergent concepts within the data were developed. Initial coding and theme development were conducted by LH and subsequently reviewed and refined by SS, who had conducted the majority of interviews and focus groups and reviewed all transcripts for accuracy. Inductive codes were grouped into sub-themes, which were iteratively reviewed and integrated with the deductive framework to form a cohesive thematic structure that balanced participant-driven insights with the study’s predetermined focus areas.
Interviews were conducted with participants from each participant group until thematic saturation was reached. While concept of saturation is debated in qualitative research [13, 14], it is generally understood as the point at which additional data no longer yield novel insights or themes. In this study, saturation was reached when no further themes arose within each participant group. Ongoing, iterative analysis conducted alongside data collection allowed researchers to monitor the recurrence of ideas in line with the study’s aim.”
24.
There is no discussion on intercoder reliability or data saturation, which are essential to assess thematic robustness.
Response: We have added the following on interrater reliability. We have included information about data saturation. Please also see the response to Reviewer 2 in Point 8.
25.
While the co-design approach is well-articulated, the study lacks an ablation or comparative analysis—e.g., comparing participant groups or evaluating different versions of the model through iteration—to substantiate the service design's efficacy.
Response: Thank you for this comment. This paper is focussed on the co-design aspect of the service. We are describing the service model based on this co-design in a separate manuscript. The efficacy of the service is currently being evaluated during the pilot of Luminos and will be described in a separate paper once it is complete. As the reviewer states, the model has been developed iteratively and has improved via a constant feedback loop. This will be reflected in the evaluation manuscript.
26.
The manuscript would benefit from a clearer articulation of exclusion/inclusion criteria for participant recruitment
Response: We have included participant exclusion criteria in the Participants section:
“Participants were excluded from participating if they were actively suicidal with intent or a plan (as assessed by the clinical judgement of the research team), appeared to be under the influence of alcohol or other drugs at the time of the focus group or interview, or were unable to provide informed consent.”
27.
More explicit detailing of how cultural safety principles were integrated beyond listing stakeholder categories.
Response: We have added this paragraph in Procedure section:
“Cultural safety principles were embedded throughout the research design and engagement processes. Specific Aboriginal community-controlled services were engaged as stakeholders, and a tailored interview guide was developed for Aboriginal participants to ensure that questions were contextually appropriate and respectful. All focus groups and interviews were conducted by experienced facilitators with research or clinical expertise in youth mental health, who had completed ASIST suicide intervention training and Aboriginal Cultural Awareness training. These measures were implemented to foster safe and inclusive spaces, supporting culturally responsive engagement in the co-design of the youth suicide service model.”
28.
Importantly, while the service has been implemented, no preliminary evaluation results or implementation fidelity data are provided, limiting the ability to assess potential clinical impact.
Response: As the evaluation is still in progress, it is premature to report on service usage or demographic data at this stage. This paper is focused specifically on the co-design process. Findings related to real-world implementation, including accommodation numbers, age breakdowns, and usage patterns, will be presented in a separate paper upon completion of data collection. To avoid any confusion, we have removed the sentence indicating that the service has already been implemented.
29.
A future direction and limitations section should be added to critically reflect on scalability, sustainability, and potential risks of peer-led models in high-risk populations.
Response: We have now included paragraphs discussing limitations and future directions and added more information on these factors as requested through comprehensive revision of the Discussion section.
Round 2
Reviewer 1 Report
Comments and Suggestions for Authors
The authors have extensively reviewed the manuscript. I have one minor suggestion, though. The discussion section has been expanded, yet most of the discussion relies primarily on the authors’ own statements, with few references. I recommend that authors may add more citations so as to better situate the contribution of this work within the prior studies.
Author Response
Comment 1: The discussion section has been expanded, yet most of the discussion relies primarily on the authors’ own statements, with few references. I recommend that authors may add more citations so as to better situate the contribution of this work within the prior studies.
Response: This has been done as requested.
Reviewer 3 Report
Comments and Suggestions for Authors
The authors have completely addressed all my comments, and I have no further concerns. Therefore, I recommend accepting the paper.
Author Response
Thank you